# Regret Bounds for Satisficing in Multi-Armed Bandit Problems

## Abstract

This paper considers the objective of *satisficing* in multi-armed bandit problems. Instead of aiming to find an optimal arm, the learner is content with an arm whose reward is above a given satisfaction level. We provide algorithms and analysis for the realizable case when such a satisficing arm exists as well as for the general case when this may not be the case. Introducing the notion of *satisficing regret*, our main result shows that in the general case it is possible to obtain constant satisficing regret when there is a satisficing arm (thereby correcting a contrary claim in the literature), while standard logarithmic regret bounds can be re-established otherwise. Experiments illustrate that our algorithm is not only superior to standard algorithms in the satisficing setting, but also works well in the classic bandit setting.

## 1 Introduction

One of the reasons why reinforcement learning (RL) is in general difficult is that finding an *optimal* policy in general requires a lot of exploration. In practice however, we are often happy to perform a task just good enough. For example, when driving to work we will be content with a strategy that will let us arrive just in time, while the computation of a policy that is 'optimal' in some sense (e.g., along the shortest route, or as fast as possible) may be prohibitive. Accordingly, it is to be expected that when considering a *satisficing* objective aiming to find a solution that is above a certain satisfaction level it is possible to learn a respective policy much faster.

Investigating the multi-armed bandit (MAB) setting, in this paper we introduce the notion of *satisficing regret* that measures the loss with respect to a given satisfaction level $S$. We first consider the realizable case, where this level can be satisfied, that is, there is at least one arm whose expected reward is above the satisfaction level. In this setting, quite a simple algorithm can be shown to have constant satisficing regret (i.e., no dependence on the horizon $T$). For the general setting we provide an algorithm that is able to extend this result, giving constant satisficing regret in the realizable case, while obtaining logarithmic bounds on the ordinary regret with respect to the optimal arm as for classic MAB algorithms such as UCB1 (Auer et al., 2002). Experiments not only confirm our theoretical findings but also show that our algorithm is competitive even in the standard setting.

### 1.1 Setting

We consider the standard multi-armed bandit (MAB) setting with a set of $K$ arms, in the following denoted as $[\![1, K]\!] := \{1, 2, \dots, K\}$. In discrete time steps $t = 1, 2, \dots$ the learner picks an arm $A_t = i$ from $[\![1, K]\!]$ and observes a random reward $r_t$ drawn from a fixed reward distribution specific to the chosen arm $i$ with mean $\mu_i$. In the following we assume that the reward distributions for each arm are sub-Gaussian. This is e.g. guaranteed when the reward distributions are bounded, which is a common assumption in the bandit setting.

The usual performance measure for a learning algorithm in the MAB setting is the *(pseudo-)regret* after $T$ steps, defined as

$$R_T := \sum_{t=1}^{T} \left( \mu_* - \mathbb{E}[\mu_{A_t}] \right),$$

where $\mu_* := \max_i \mu_i$ is the maximal mean reward over all arms.

In the satisficing setting however, we only care about whether an arm with mean reward $\geqslant S$ is chosen, where $S$ is the level of satisfaction we aim at. Accordingly, we modify the classic notion of regret and consider what we call the *satisficing (pseudo-)regret* with respect to $S$ (short *S-regret*) defined as

$$R_T^S := \sum_{t=1}^{T} \max \left\{ S - \mathbb{E}[\mu_{A_t}], 0 \right\}.$$

This definition reflects that we are happy with any arm having mean reward $\geqslant S$ and that there is no benefit in overfulfilling the given satisfaction level $S$. Note that the $S$-regret will be linear in $T$ whenever there is no satisficing arm with mean reward $\geqslant S$, that is, if $\mu_* < S$. As will be discussed below, $S$-regret is a special case of the notion of *expected satisficing regret* as considered in a more general Bayesian setting introduced by Reverdy et al. (2017).

## 1.2   Related Work

While there are some connections to multi-criterion RL (Roijers et al., 2013), there is hardly any literature on satisficing in RL, with a few exceptions for the MAB setting that we also consider. Kohno and Takahashi (2017) and Tamatsukuri and Takahashi (2019) propose simple index policies, which are experimentally evaluated. Tamatsukuri and Takahashi (2019) also show that the suggested algorithm converges to a satisficing arm and that the regret is finite if the satisfaction level is chosen to be between the reward of the best and the second-best arm.

Reverdy et al. (2017) consider a more general Bayesian setting, which also considers the learner's belief that some arm is satisficing. The notion of *expected satisficing regret* is introduced that measures the loss over all steps where a non-satisficing arm is chosen and the learner's degree of belief in the chosen arm was below some level $\delta \in [0, 1]$. For $\delta = 0$ this coincides with our notion of *satisficing regret* as defined above. Reverdy et al. (2017) present various bounds on the expected satisficing regret, including lower bounds as well as upper bounds for problems with Gaussian reward distributions when using adaptations of the UCL algorithm (Reverdy et al., 2014). The given bounds for the case $\delta = 0$ that correspond to our setting will be discussed in Section 2 below.

A line of reseach that pursues similar ideas as our setting of satisficing is that of *conservative bandits*. Here the learner has an arm at her disposal that provides a baseline level (similar to our satisfaction level) one would not like to fall below, while trying to converge to an optimal arm. Thus Wu et al. (2016) present an algorithm that on the one hand with high probability stays above the baseline level at all time steps (with a certain amount of allowed error $\alpha$) and on the other hand has regret bounded similar to standard bandit algorithms (but with an additional dependence on $\alpha$).

Merlis and Mannor (2021) consider a related notion of so-called *lenient regret* that considers the loss with respect to $\mu_* - \varepsilon$ for a parameter $\varepsilon > 0$ that specifies the allowed deviation from the optimal mean reward $\mu_*$. The definition of lenient regret formally depends on a so-called $\varepsilon$-gap function. When choosing this function to be the hinge loss, lenient regret corresponds to $S$-regret when choosing $S := \mu^* - \varepsilon$. Merlis and Mannor (2021) show asymptotic upper bounds on the lenient regret for a version of Thompson sampling (Thompson, 1933) that match a given lower bound. When $\mu_* > 1 - \varepsilon$ the lenient regret turns out to be constant. This resembles the results we have for $S$-regret, which in Theorems 1 and 2 below is shown to be constant when there is a satisficing arm. However, the results are not equivalent: The lenient regret is with respect to the value $\mu^* - \varepsilon$ and constant regret is only obtained when $\mu^*$ itself is $\varepsilon$-close to the theoretical maximum reward 1. On the other hand, considering an absolute satisfaction level $S$ we obtain constant $S$-regret whenever $\mu^* > S$. This holds in particular when $S$ is chosen to be $\mu^* - \varepsilon$ and without further assumptions on $\mu^*$.

Russo and Roy (2018) consider satisficing in a setting with discounted rewards and provide respective bounds on the expected discounted regret for a satisficing variant of Thompson sampling.

Also related to our paper, Kano et al. (2019) consider the problem of identifying *all* arms above a given satisfaction level and derive sample complexity bounds for the pure-exploration setting with fixed confidence. Related sample complexity bounds can be found in (Mason et al., 2020) for identification of all $\varepsilon$-good arms. Closer to our setting is the problem of identifying an arbitrary arm among the top $m$ arms, for which sample complexity bounds are derived by Chaudhuri and Kalyanakrishnan (2017). A follow-up paper (Chaudhuri and Kalyanakrishnan, 2019) considers the sample complexity of the more general problem of identification of any $k$ of the best $m$ arms. None of these latter investigations however considers the online learning setting with regret as performance measure as we do. Note that an algorithm for pure exploration (Audibert et al., 2010) after any number of steps with high probability will identify an optimal or at least a satisficing arm. However, subsequent exploitation will always give linear regret due to the small but positive error probability so that a simple approach of first exploring and then exploiting does not work well in general.

## 2 The Realizable Case

We start with the *realizable case* when $\mu_* > S$. The main goal of this section is to show that suitable algorithms will have just constant $S$-regret in this case. Note that this does not hold for standard algorithms like UCB1 (Auer et al., 2002). Lower bounds show that these algorithms will choose a suboptimal arm $i$ for $\Omega\left(\frac{logT}{(\mu^*-\mu_i)^2}\right)$ times. This of course also holds for any arm below the satisfaction level $S$ giving a contribution to the overall $S$-regret of $\Omega\left(\frac{(S-\mu_i)\log T}{(\mu^*-\mu_i)^2}\right)$.

### 2.1 Simple Algorithm

We start with the simple algorithm SIMPLE-SAT shown as Algorithm 1. It plays the empirical best arm so far if its empirical mean reward is $\geqslant S$ and explores uniformly at random otherwise. In the following, the empirical reward for arm $i$ available at step $t$ (i.e., *before* choosing the arm $A_t$) is denoted by $\hat{\mu}_i(t)$.

---

**Algorithm 1**: SIMPLE-SAT (Simple Algorithm for Satisficing in the Realizable Case)

---

**Require:** $K$, $S$
 1: Play each arm once, i.e., for time steps $t = 1, \ldots, K$ play arm $A_t = t$.
 2: **for** time steps $t = K + 1, \ldots$ **do**
 3:     **if** $\exists i \, \hat{\mu}_i(t) \geqslant S$ **then**
 4:         Play $A_t \leftarrow \arg\max_{i \in [\![1,K]\!]} \hat{\mu}_i(t)$.
 5:     **else**
 6:         Choose $A_t$ uniformly at random from $[\![1, K]\!]$.
 7:     **end if**
 8: **end for**

---

Analogously to the ordinary MAB setting where the gaps $\Delta_i := \mu_* - \mu_i$ to the optimal arm appear in bounds on the (classic) regret, when satisficing the gaps $\Delta_i^S = S - \mu_i$ for non-satisficing arms as well as $|\Delta_*^S| = \mu_* - S$ are important parameters describing the difficulty of the problem. Indeed, one can show the following bound on the $S$-regret.

**Theorem 1.** *If $S < \mu_*$ then* SIMPLE-SAT *satisfies for all $T \geqslant 1$,*

$$R_T^S \leqslant \sum_{i:\Delta_i^S > 0} \left( \Delta_i^S + \frac{2}{\Delta_i^S} + \frac{2\Delta_i^S}{|\Delta_*^S|^2} \right).$$

For the proof we shall need the following result that follows by our assumption of sub-Gaussianity and a Chernoff bound.

**Lemma 1.** *Let $\hat{\mu}_{i,n}$ be an empirical estimate for $\mu_i$ computed from $n$ samples. Then for all $\varepsilon > 0$ and each $i \in [\![1, K]\!]$,*

$$\mathbb{P}(\hat{\mu}_{i,n} \geqslant \mu_i + \varepsilon) \leqslant \exp(-\tfrac{n\varepsilon^2}{2}),$$
$$\mathbb{P}(\hat{\mu}_{i,n} \leqslant \mu_i - \varepsilon) \leqslant \exp(-\tfrac{n\varepsilon^2}{2}).$$

*Proof of Theorem 1.* Let $i$ be the index of a non-satisficing arm. In the following we decompose the event that arm $i$ is chosen at some step $t$. To do that we introduce the event $Z_t := \{\forall j \in [\![1, K]\!], \hat{\mu}_j(t) < S\}$ that all arms have empirical estimates below $S$, when the algorithm chooses an arm randomly according to line 6 of the algorithm. Then we have

$$\{A_t = i\} \subset \{t = i\} \cup \{A_t = i, Z_t^c\} \cup \{A_t = i, Z_t\}. \tag{1}$$

For the first two events we have

$$\sum_{t=1}^{T} \mathbb{P}(t = i) \leqslant 1 \tag{2}$$

and

$$\begin{aligned}
\sum_{t=1}^{T} \mathbb{P}(A_t = i, Z_t^c) &\leqslant \sum_{t=1}^{T} \mathbb{P}\big(A_t = i, \hat{\mu}_i(t) \geqslant S\big) = \sum_{t=1}^{T} \mathbb{P}\big(A_t = i, \hat{\mu}_i(t) \geqslant \mu_i + \Delta_i^S\big) \\
&\leqslant \sum_{n=1}^{T} \mathbb{P}(\hat{\mu}_{i,n} \geqslant \mu_i + \Delta_i^S) \\
&\leqslant \sum_{n=1}^{T} \exp\big(-\tfrac{n(\Delta_i^S)^2}{2}\big) \\
&\leqslant \frac{e^{-\frac{(\Delta_i^S)^2}{2}}}{1 - e^{-\frac{(\Delta_i^S)^2}{2}}} \\
&\leqslant \frac{2}{(\Delta_i^S)^2}. \tag{3}
\end{aligned}$$

Rewriting the probability of the third event in eq. 1, using $*$ to refer to an arbitrary optimal arm, we obtain

$$\begin{aligned}
\mathbb{P}(A_t = i, Z_t) = \mathbb{P}(A_t = i | Z_t)\,\mathbb{P}(Z_t) &= \tfrac{1}{K} \cdot \mathbb{P}(Z_t) \\
&= \mathbb{P}(A_t = * | Z_t)\,\mathbb{P}(Z_t) = \mathbb{P}(A_t = *, Z_t).
\end{aligned}$$

Now summing over the time steps up to $T$ yields

$$\begin{aligned}
\sum_{t=1}^{T} \mathbb{P}(A_t = i, Z_t) &= \sum_{t=1}^{T} \mathbb{P}(A_t = *, Z_t) \\
&\leqslant \sum_{t=1}^{T} \mathbb{P}\big(A_t = *, \hat{\mu}_*(t) \leqslant S\big) = \mathbb{E}\Big(\sum_{t=1}^{T} \mathbb{1}\big\{A_t = *, \hat{\mu}_*(t) \leqslant S\big\}\Big) \\
&\leqslant \mathbb{E}\Big(\sum_{n=1}^{T} \mathbb{1}\{\hat{\mu}_{*,n} \leqslant S\}\Big) = \sum_{n=1}^{T} \mathbb{P}(\hat{\mu}_{*,n} \leqslant S) \\
&= \sum_{n=1}^{T} \mathbb{P}\big(\hat{\mu}_{*,n} \leqslant \mu_* - |\Delta_*^S|\big) \\
&\leqslant \sum_{n=1}^{T} \exp\big(-\tfrac{n|\Delta_*^S|^2}{2}\big) \\
&\leqslant \frac{2}{|\Delta_*^S|^2}. \tag{4}
\end{aligned}$$

Finally writing

$$n_i(T) = \sum_{t=1}^{T} \mathbb{1}\{A_t = i\}$$

for the number of times arm $i$ was pulled up to step $T$, we can combine eqs. 1–4 to obtain

$$
\begin{aligned}
R_T^S &= \sum_{i:\Delta_i^S>0} \Delta_i^S \, \mathbb{E}(n_i(T)) \\
&= \sum_{i:\Delta_i^S>0} \Delta_i^S \sum_{t=1}^{T} \mathbb{P}(A_t = i) \\
&\leqslant \sum_{i:\Delta_i^S>0} \Delta_i^S \left(1 + \frac{2}{(\Delta_i^S)^2} + \frac{2}{|\Delta_*^S|^2}\right) \\
&= \sum_{i:\Delta_i^S>0} \left(\Delta_i^S + \frac{2}{\Delta_i^S} + \frac{2\Delta_i^S}{|\Delta_*^S|^2}\right). \qquad\qquad \square
\end{aligned}
$$

The algorithm as well as the analysis are adaptations from (Bubeck et al., 2013) where ordinary regret bounds for the MAB setting are considered under the assumption that the learner knows the value of $\mu_*$ as well as (a bound on) the gap $\Delta$ between the optimal and the best suboptimal arm.[1] The crucial insight is that what is actually needed in order to apply algorithm and analysis of Bubeck et al. (2013) is to have a reference value $\mu$ that separates the optimal from suboptimal arms, that is, $\mu_* > \mu > \mu_i$ for all suboptimal arms $i$. In our case this reference value is given by the satisfaction level $S$, which in the realizable case separates the good arms from the bad ones. Note that for this we need to have $S < \mu_*$, so that we do not get constant regret when $S = \mu_*$.

*Remarks.* (i) Concerning the quality of the upper bound of Theorem 1, standard lower bounds imply that for an arm $i$ with unknown mean reward $\mu_i < S$ one needs $\sim (\Delta_i^S)^{-2}$ samples to determine that arm $i$ is not satisfying. As the respective $S$-regret for choosing arm $i$ is $\Delta_i^S$, this gives a lower bound on the regret of order $(\Delta_i^S)^{-1}$. Similarly, in order to be sure that the optimal arm with mean reward $\mu_* > S$ is satisfying it takes $\sim (\Delta_*^S)^{-2}$ samples. Although sampling $\mu*$ does not incur any regret it is intuitive that the problem becomes more difficult for smaller $\Delta_*^S$ so that it is intuitive that this term appears in the bound of Theorem 1. However, a matching lower bound is only available for special cases such as when $\Delta_i^S = \Delta_*^S$ for all not satisfying arms $i$. As neither $\Delta_i^S$ nor $\Delta_*^S$ is known to the learner, it is not obvious how our simple algorithm and the respective upper bound could be improved either. We conjecture that the upper bound of Theorem 1 is basically optimal.

(ii) The constant regret bound of Theorem 1 not only improves over the logarithmic bounds given by Reverdy et al. (2017) for a variant of the UCL algorithm (Reverdy et al., 2014) that picks an arbitrary arm with UCL-index above $S$ (instead of an arm with maximal index). Our bound also is not consistent with a claimed lower bound that is also logarithmic in the horizon (not mentioned in the corrections of Reverdy et al., 2021). This bound is obtained by application of a lower bound for the *multiple play* setting (Anantharam et al., 1987), where at each step $m$ arms are chosen by the learner, who hence has to identify the $m$ best arms. The given proof chooses $m$ to be all arms above the given satisfaction level $S$. However, the lower bound is obviously not directly applicable to the satisficing setting: not *all* arms above the satisfaction level have to be found, but a single one is sufficient.

Bubeck et al. (2013) also provide another algorithm with a more refined approach for exploration, using a potential function instead of a uniform probability distribution over the arms. We note that a respective adaptation of algorithm and analysis to the satisficing setting can be done in a quite straightforward way.

---

[1]As has been shown in the meantime, knowledge of $\mu_*$ is sufficient for obtaining constant bounds on the regret (Garivier et al., 2019).

## 3 The General Case

Now let us consider the general case where it is not guaranteed that the chosen satisfaction level $S$ is realizable, that is, it may happen that $S > \mu_*$. Then unlike in the realizable case the satisfaction level $S$ does not give the learner any useful information so that we cannot hope to perform better than in an ordinary MAB setting. Obviously the $S$-regret will be linear, but we can still aim at getting bounds on the (classic) regret. On the other hand, if there is at least one arm above the satisfaction level $S$, we would like to re-establish constant bounds on the $S$-regret as in the realizable case.

For the general setting we propose the SAT-UCB scheme shown as Algorithm 3. SAT-UCB exploits when there is an arm with empirical mean above $S$ (cf. line 4 of the algorithm) and explores otherwise. The exploration takes into account a UCB value similar to the classical index suggested for the UCB1 algorithm of (Auer et al., 2002), that is,

$$\text{UCB}_i(t) := \hat{\mu}_i(t) + \beta_i(t),, \tag{5}$$

where

$$\beta_i(t) = \sqrt{\tfrac{2\log(f(t))}{n_i(t-1)}}$$

with $f(t) = 1 + t\log^2(t)$. If there is at least one arm with UCB-value above $S$ then SAT-UCB chooses such an arm uniformly at random, which makes sure that all promising arms are explored sufficiently to decide whether they are satisficing. Otherwise, if all arms have UCB-value below $S$, the algorithm chooses an arm according to UCB1, that is, an arm $i$ maximizing $\text{UCB}_i$. This guarantees that the algorithm performs similar to UCB1 when there is no satisficing arm.

---

**Algorithm 3**: SAT-UCB Scheme for Satisficing in the General Case

**Require:** $K, S$
 1: Play each arm once, i.e., for time steps $t = 1, \ldots, K$ play arm $A_t = t$.
 2: **for** time steps $t = K+1, \ldots$ **do**
 3:     **if** $\exists i \, \hat{\mu}_i(t) \geqslant S$ **then**
 4:         Choose an arbitrary $A_t$ from $\{i \,|\, \hat{\mu}_i(t) \geqslant S\}$.
 5:     **else if** $\exists i \, \text{UCB}_i(t) \geqslant S$ **then**
 6:         Choose $A_t$ uniformly at random from $\{i \,|\, \text{UCB}_i(t) \geqslant S\}$.
 7:     **else**
 8:         Play arm $A_t \in \underset{i\in[\![1,K]\!]}{\arg\max} \text{UCB}_i(t)$ .
 9:     **end if**
10: **end for**

---

For step 4 in SAT-UCB different concrete instantiations of the exploitation step are possible. In Section 4 we will consider different sub-algorithms for choosing an arm from $\{i \,|\, \hat{\mu}_i(t) \geqslant S\}$. The following two theorems are independent of the selected exploitation sub-algorithm and show that SAT-UCB achieves constant $S$-regret if $\mu_* > S$, while the regret is bounded as for UCB1 (Auer et al., 2002) otherwise.

**Theorem 2.** *If $\mu_* > S$ then SAT-UCB satisfies for all $T \geqslant 1$,*

$$R_T^S \leqslant \sum_{i:\Delta_i^S>0} \left( \Delta_i^S + \frac{2}{\Delta_i^S} + \frac{7\Delta_i^S}{|\Delta_*^S|^2} \right).$$

*Proof.* As before we write the $S$-regret as

$$R_T^S = \sum_{i:\Delta_i^S>0}^{k} \mathbb{E}(n_i(T)) \, \Delta_i^S$$

and proceed bounding $\mathbb{E}(n_i(T)) = \sum_{t=1}^{T} \mathbb{P}(A_t = i)$ for all non-satisfying arms $i$. Thus let $i$ be the index of a non-satisfying arm. Let $Z_t := \{\forall j \in [\![1, K]\!], \hat{\mu}_j(t) < S\}$ be again the event that all arms have empirical values below the satisfaction level. Then we can decompose the event $\{A_t = i\}$ as

$$
\begin{aligned}
\{A_t = i\} \subset &\{t = i\} \cup \{A_t = i, \hat{\mu}_i(t) \geqslant S, t > K\} \\
&\cup \{A_t = i, \mathrm{UCB}_i(t) \geqslant S, \mathrm{UCB}_*(t) \geqslant S, t > K, Z_t\} \\
&\cup \{A_t = i, \mathrm{UCB}_*(t) < S, t > K, Z_t\}.
\end{aligned}
\tag{6}
$$

For the first two events we have

$$
\sum_{t=1}^{T} \mathbb{P}(t = i) \leqslant 1
\tag{7}
$$

and analogously to eq. 3

$$
\begin{aligned}
\sum_{t=1}^{T} \mathbb{P}(A_t = i, \, \hat{\mu}_i(t) \geqslant S, t > K) &\leqslant \sum_{t=1}^{T} \mathbb{P}(A_t = i, \hat{\mu}_i(t) \geqslant \mu_i + \Delta_i^S) \\
&\leqslant \sum_{n=1}^{T} \mathbb{P}(\hat{\mu}_{i,n} \geqslant \mu_i + \Delta_i^S) \\
&\leqslant \frac{2}{(\Delta_i^S)^2} \, .
\end{aligned}
\tag{8}
$$

For the probability of the third event we have

$$
\begin{aligned}
\sum_{t=1}^{T} &\mathbb{P}(A_t = i, \mathrm{UCB}_i(t) \geqslant S, \mathrm{UCB}_*(t) \geqslant S, t > K, Z_t) \\
&= \sum_{t=1}^{T} \mathbb{P}(A_t = *, \mathrm{UCB}_i(t) \geqslant S, \mathrm{UCB}_*(t) \geqslant S, t > K, Z_t) \\
&\leqslant \sum_{t=1}^{T} \mathbb{P}(A_t = *, Z_t) \leqslant \sum_{t=1}^{T} \mathbb{P}(A_t = *, \hat{\mu}_*(t) \leqslant S) = \mathbb{E}\Big( \sum_{t=1}^{T} \mathbb{1}\{A_t = *, \hat{\mu}_*(t) \leqslant S\} \Big) \\
&\leqslant \mathbb{E}\Big( \sum_{n=1}^{T} \mathbb{1}\{\hat{\mu}_{*,n} \leqslant S\} \Big) = \sum_{n=1}^{T} \mathbb{P}(\hat{\mu}_{*,n} \leqslant S) = \sum_{n=1}^{T} \mathbb{P}(\hat{\mu}_{*,n} \leqslant \mu_* - |\Delta_*^S|) \\
&\leqslant \sum_{n=1}^{T} \exp\big( -\tfrac{n|\Delta_*^S|^2}{2} \big) \\
&\leqslant \frac{2}{|\Delta_*^S|^2}.
\end{aligned}
\tag{9}
$$

Finally, the probability of the last event of eq. 6 is upper bounded by

$$
\begin{aligned}
\sum_{t=1}^{T} \mathbb{P}(\mathrm{UCB}_*(t) < S) &= \sum_{t=1}^{T} \mathbb{P}(\hat{\mu}_*(t) < \mu_* - (|\Delta_*^S| + \beta_*(t))) \\
&\leqslant \sum_{t=1}^{T} \sum_{n=1}^{t} \mathbb{P}\left( \hat{\mu}_{*,n} < \mu_* - \Big( |\Delta_*^S| + \sqrt{\tfrac{2\log(f(t))}{n}} \Big) \right) \\
&\leqslant \sum_{t=1}^{T} \sum_{n=1}^{t} \frac{1}{f(t)} \exp\big( -\tfrac{n|\Delta_*^S|^2}{2} \big) \leqslant \frac{2}{|\Delta_*^S|^2} \sum_{t=1}^{T} \frac{1}{f(t)} \\
&\leqslant \frac{5}{|\Delta_*^S|^2} \, ,
\end{aligned}
\tag{10}
$$

where the last inequality is obtained by observing that $\sum_{t=1}^{T} \frac{1}{f(t)} \leqslant 1 + \sum_{t=2}^{T} \frac{1}{t \log^2(t)}$ and then bounding the sum with an integral.

Finally, by putting everything together, we obtain from equations eqs. 6– 10 the claimed result

$$R_T^S = \sum_{i:\Delta_i^S > 0} \Delta_i^S \, \mathbb{E}(n_i(T)) \leqslant \sum_{i:\Delta_i^S > 0} \left( \Delta_i^S + \frac{2}{\Delta_i^S} + \frac{7\Delta_i^S}{|\Delta_*^S|^2} \right).$$

$\square$

**Theorem 3.** *If $\mu_* \leqslant S$ then SAT-UCB satisfies for all $T \geqslant 1$*

$$R_T \leqslant \sum_{i:\Delta_i > 0} \inf_{\varepsilon \in (0, \Delta_i)} \Delta_i \left( 1 + \frac{5}{\varepsilon^2} + \frac{2(\log f(T) + \sqrt{\pi \log f(T)} + 1)}{(\Delta_i - \varepsilon)^2} \right). \tag{11}$$

*Furthermore,*

$$\limsup_{T \to \infty} \frac{R_T}{\log(T)} \leqslant \sum_{i:\Delta_i > 0} \frac{2}{\Delta_i}. \tag{12}$$

*Thus, for a constant $C > 0$ it holds that*

$$R_T \leqslant C \sum_{i:\Delta_i > 0} \left( \Delta_i + \frac{\log(T)}{\Delta_i} \right).$$

*Proof.* The proof can be reduced to the derivation of the regret bounds for UCB1 as given in Theorem 8.1 of Lattimore and Szepesvári (2020). We start with the standard regret decomposition

$$R_T = \sum_{i:\Delta_i > 0} \mathbb{E}\left(n_i(T)\right) \Delta_i.$$

In the following, we bound for each suboptimal arm $i$ the number of times $n_i(T)$ it is played. By definition of the algorithm, arm $i$ is chosen after step $K$ only if either

$$\hat{\mu}_i(t) + \beta_i(t) \geqslant \hat{\mu}_*(t) + \beta_*(t) \quad \text{or} \quad \hat{\mu}_i(t) + \beta_i(t) \geqslant S.$$

(Note that the case $\hat{\mu}_i(t) \geqslant S$ is subsumed by the second event.) Accordingly, we can decompose the event $A_t = i$ using some arbitrary but fixed $\varepsilon \in (0, \Delta_i)$ as

$$\begin{aligned} \{A_t = i\} \subseteq & \big\{ A_t = i \text{ and } \hat{\mu}_*(t) + \beta_*(t) \leqslant \mu_* - \varepsilon \big\} \\ & \cup \big\{ A_t = i \text{ and } \hat{\mu}_*(t) + \beta_*(t) \geqslant \mu_* - \varepsilon \big\} \\ \subseteq & \big\{ \hat{\mu}_*(t) + \beta_*(t) \leqslant \mu_* - \varepsilon \big\} \\ & \cup \big\{ A_t = i \text{ and } \hat{\mu}_i(t) + \beta_i(t) \geqslant \hat{\mu}_*(t) + \beta_*(t) \geqslant \mu_* - \varepsilon \big\} \\ & \cup \big\{ A_t = i \text{ and } \hat{\mu}_*(t) + \beta_*(t) \geqslant \mu_* - \varepsilon \text{ and } \hat{\mu}_i(t) + \beta_i(t) \geqslant S \big\} \\ \subseteq & \big\{ \hat{\mu}_*(t) + \beta_*(t) \leqslant \mu_* - \varepsilon \big\} \\ & \cup \big\{ A_t = i \text{ and } \hat{\mu}_i(t) + \beta_i(t) \geqslant \mu_* - \varepsilon \big\}, \end{aligned}$$

where the last inclusion is due to the assumption that $\mu_* \leqslant S$. It follows that

$$n_i(T) \leqslant \sum_{t=1}^{T} \mathbb{1}\big\{ \hat{\mu}_*(t) + \beta_*(t) \leqslant \mu_1 - \varepsilon \big\} + \sum_{t=1}^{T} \mathbb{1}\big\{ A_t = i \text{ and } \hat{\mu}_i(t) + \beta_i(t) \geqslant \mu_* - \varepsilon \big\}.$$

The obtained decomposition is the same as the one in the proof of Theorem 8.1 from (Lattimore and Szepesvári, 2020) and the very same arguments can be used to finish the proof of eq. 11. The second part of the theorem, that is eq. 12, follows by choosing $\varepsilon = \log^{-1/4}(T)$ and taking the limit as $T$ tends to infinity. $\square$

## 4 Experiments

We compared SAT-UCB to other bandit algorithms in order to show that the latter keep accumulating $S$-regret, while SAT-UCB sticks to a satisficing arm after finite time, thus confirming the results of Theorem 2. We also investigated the behavior of SAT-UCB in the not realizable case with different values for the chosen satisfaction level $S$ and did experiments with a slightly modified version SAT-UCB$^+$ introduced below in Section 4.2.

### 4.1 Exploitation in Sat-UCB

As already mentioned, we investigated different sub-algorithms for exploitation in step 4 of SAT-UCB. Obvious choices for this exploitation step are e.g. selecting the arm with maximal empirical mean reward or using UCB1 to choose among the arms with empirical mean reward above $S$.

However, the following more refined approach for exploitation empirically worked best. In addition to the UCB value for each arm we define an analogous lower confidence bound value

$$\text{LCB}_i(t) := \hat{\mu}_i(t) - \beta_i(t). \tag{13}$$

Then for any arm $i$ with empirical mean above $S$ we consider the confidence interval $[LCB_i, UCB_i]$ and then choose the arm for which the largest share of this confidence interval is above the satisfaction level $S$. That is, step 4 of SAT-UCB chooses an arm from

$$\underset{i \in [\![1,K]\!]}{\operatorname{argmax}} \left\{ \frac{\text{UCB}_i(t) - \max\{S,\, \text{LCB}_i(t)\}}{\beta_i(t)} \right\}. \tag{14}$$

The intuition behind this choice is that an arm whose confidence interval ist mostly above $S$ will most likely have actual mean above $S$. In the following experiments, *SAT-UCB* always refers to SAT-UCB employing this *confidence fraction* index as exploitation sub-algorithm.

### 4.2 Sat-UCB$^+$: Modified Exploration in Sat-UCB

Concerning exploration we also considered a simplified version of SAT-UCB which does not use random exploration and instead always plays UCB1 when there is no empirically satisficing arm. For the sake of completeness, this modification SAT-UCB$^+$ is shown as Algorithm 4.

---

**Algorithm 4**: SAT-UCB$^+$ (Experimental Simplification of SAT-UCB)

---

**Require:** $K, S$
1:  Play each arm once, i.e., for time steps $t = 1, \ldots, K$ play arm $A_t = t$.
2:  **for** time steps $t = K + 1, \ldots$ **do**
3:      **if** $\exists i \, \hat{\mu}_i(t) \geqslant S$ **then**
4:          Choose $A_t$ from $\underset{i \in [\![1,K]\!]}{\operatorname{argmax}} \left\{ \frac{\text{UCB}_i(t) - \max\{S,\, \text{LCB}_i(t)\}}{\beta_i(t)} \right\}$.
5:      **else**
6:          Play arm $A_t \in \underset{i \in [\![1,K]\!]}{\operatorname{argmax}} \text{UCB}_i(t)$ .
7:      **end if**
8:  **end for**

---

While we were not able to provide a constant bound on the $S$-regret as for the original SAT-UCB algorithm, in the experiments SAT-UCB$^+$ performed better than SAT-UCB.

### 4.3 Setup

#### 4.3.1 Settings

To illustrate the influence of the structure of the underlying bandit problem we performed experiments in the following two settings each with 20 arms and normally distributed rewards[2] with standard deviation 1:

In *Setting 1* the mean reward of each arm $i = 1, 2, \ldots, 20$ is set to $\frac{i-1}{20}$. For the satisfaction level we chose 0.8 in the realizable case resulting in four satisfying arms. For experiments in the not realizable case $S = 1$ was chosen.

In *Setting 2* the mean reward of each arm $i$ is set to $\sqrt{\frac{i}{20}}$. For the satisfaction level we chose $S = 0.92$ so that there are three satisfying arms in the realizable case. This setting is more difficult than Setting 1, as arms are closer to $S$ as well as to each other. The not realizable satisfaction level was set to 1.1.

Further experiments for a complementary very simple setting are reported in the appendix.

#### 4.3.2 Algorithms

We compared different instantiations of SAT-UCB. Beside the main variant that uses the confidence fraction index in eq. 14 for exploitation in line 4 of SAT-UCB, we also consider using UCB1 or the maximal empirical mean for choosing an arm, respectively. We also performed experiments with the modified SAT-UCB$^+$ algorithm.

For comparison, beside UCB1 (Auer et al., 2002) (choosing the same confidence intervals as for SAT-UCB) we used UCB$_\alpha$ (Degenne et al., 2019) as well as the Satisfaction in Mean Reward UCL algorithm (Reverdy et al., 2017) and the (deterministic) UCL algorithm (Reverdy et al., 2014) it is based on.

For UCB$_\alpha$ that aims to combine regret minimization and best arm identification we chose confidence $\delta = 0.001$ and $\alpha = 1$ to focus on exploitation. We also performed a few experiments with $\alpha > 1$ that confirmed that in this case the algorithm explores more and accumulates more $S$-regret with increasing $\alpha$. The original UCB$_\alpha$ is given for just two arms and stops when the optimal arm has been identifed. In our case with an arbitrary number of arms we eliminate arms that are identified as suboptimal with respect to same criterion as suggested by Degenne et al. (2019).

For Satisfaction in Mean Reward UCL we considered various ways of how to choose arms from the *eligible set* (cf. eq. 28 of Reverdy et al., 2017), as this step is not specified in the original paper. Experimentally it worked best to select at any step $t$ *each* arm from the eligible set once but increase the time step counter just by 1, independent of how many arms have been chosen at $t$. For the parameter $a$ we chose $a = 1$, which was suggested in the original paper and worked best experimentally, although Reverdy et al. (2021) state that for the theoretical results to hold one should have $a > \frac{4}{3}$.

### 4.4 Results

#### 4.4.1 Realizable Case

We started with comparing SAT-UCB to UCB1 in Setting 1. Figure 1a depicts a showcase run in Setting 1 illustrating that SAT-UCB soon focuses on a satisficing arm, while UCB1 keeps exploring. Accordingly, UCB1 suffers growing $S$-regret due to ongoing exploration of arms below the satisfaction level, cf. Figure 2 below.

Figure 1b shows a comparison of the $S$-regret of the different versions of SAT-UCB as well as the experimental modification SAT-UCB$^+$ in Setting 1. Here and in the following plots, we show the averaged values for ($S$-)regret over 500 runs and error bars indicate a 90% confidence interval (between the 5%- and the 95%-percentile). We see that the experimental modification SAT-UCB$^+$ not only achieves the smallest $S$-regret, it also displays much smaller variance than the SAT-UCB variants. Experiments for Setting 2 give similar

---

[2]We repeated all experiments also with Bernoulli rewards, which gave similar results, which are consequently not reported here.

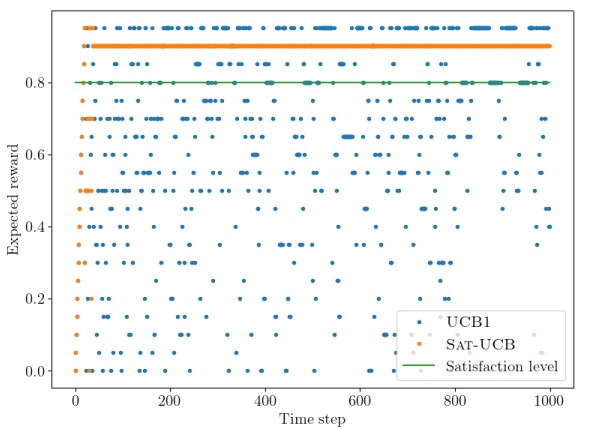 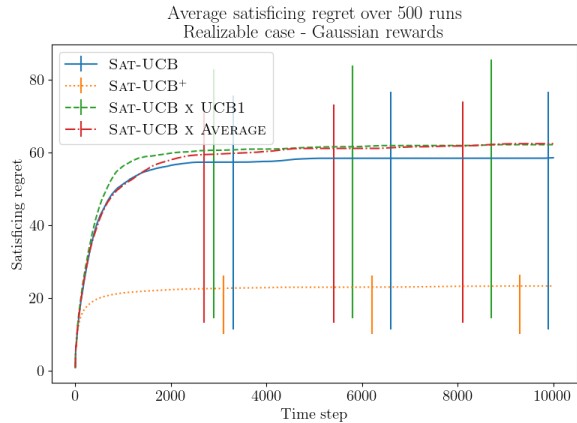

(a) Arm pulls for UCB1 and Sᴀᴛ-UCB in exemplary run.  (b) Comparing the $S$-regret for variants of Sᴀᴛ-UCB.

Figure 1: Experiments in the realizable case of Setting 1.

results (cf. Fig. 2b below). We note that for Sᴀᴛ-UCB$^+$ the confidence fraction index of eq. 14 also gives better results than using UCB1 or choosing the empirically best arm instead.

Figure 2 shows a comparison of the $S$-regret of Sᴀᴛ-UCB and Sᴀᴛ-UCB$^+$ to other algorithms. Although $S$-regret is smaller than classic regret, all algorithms except our two algorithms suffer growing regret due to ongoing exploration of arms below the satisfaction level. As expected, Sᴀᴛ-UCB gives constant regret, while surprisingly Deterministic UCL is superior to its Satisfaction in Mean Reward counterpart. Figure 2b illustrates that the regret for Sᴀᴛ-UCB is larger in Setting 2 in which the gaps of the relevant arms to the satisfaction level are smaller. In both cases Sᴀᴛ-UCB$^+$ performs best however.

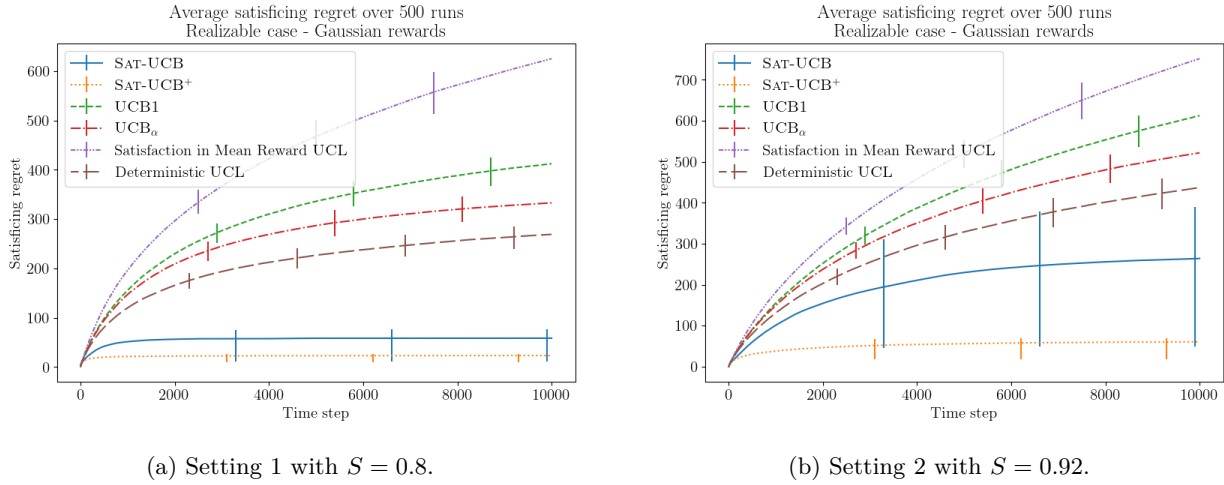

(a) Setting 1 with $S = 0.8$.  (b) Setting 2 with $S = 0.92$.

Figure 2: Comparison of $S$-regret of different algorithms in the realizable case.

### 4.4.2 Not Realizable Case

In the not realizable case Sᴀᴛ-UCB usually performed a bit worse than UCB1, cf. Figure 3a. (The respective plot for Setting 2 can be found in Figure 4a of the appendix.) In particular, in the beginning Sᴀᴛ-UCB does more (random) exploration and catches up only for larger horizon. The experimental modification Sᴀᴛ-UCB$^+$ shows the opposite behavior performing much better for small horizon before performance coincides with UCB1 and Sᴀᴛ-UCB after a higher number of steps.

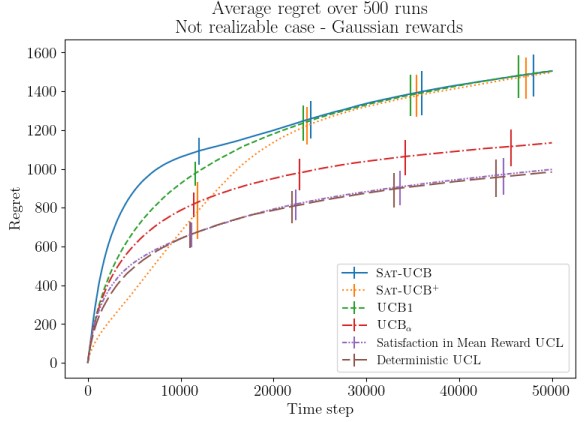
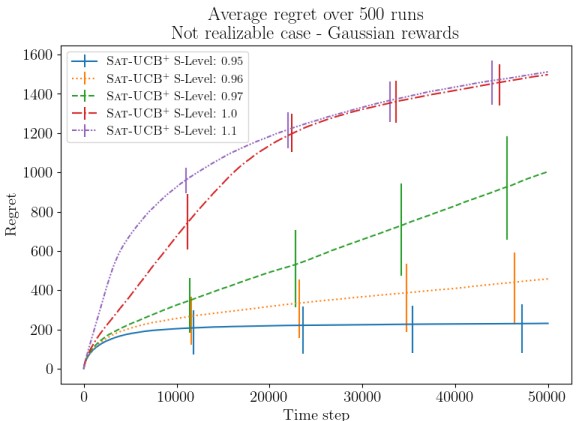

(a) Comparison of classic regret of different algorithms in the not realizable case of Setting 1 with $S = 1$.

(b) Comparison of different choices of S for SAT-UCB$^+$ in Setting 1.

Figure 3: Classic regret in the not realizable case.

Interestingly, while SAT-UCB is quite insensitive to the choice of $S$, in the not realizable case SAT-UCB$^+$ works better the closer $S$ is chosen to $\mu_*$. This is illustrated in Figure 3b. When $S$ is chosen close to $\mu_* = 1$ the regret becomes nearly constant. This behavior of SAT-UCB$^+$ can be explained as follows: When $S$ is close to $\mu^*$ this increases exploitation in case there are arms with empirical mean above $S$. With $S$ being close to $\mu^*$ it is also more likely that such arms exist. On the other hand, if there are no such arms the increased exploitation when using UCB1 (instead of random exploration as in SAT-UCB) leads to improved performance of SAT-UCB$^+$.

## 5 Conclusion

Our results for the multi-armed bandit case are just a first step in an ongoing project on satisficing in reinforcement learning. While some ideas may be used also in the general standard Markov decision process setting, it seems already not quite simple to obtain reasonable constant regret bounds in the realizable case. While it might be possible to consider each policy as an arm in an MAB setting, the resulting bounds would be linear in the number of policies and hence exponential in the number of states.

Another interesting direction of further research is satisficing with adaptive satisfaction level. While overall such an approach obviously will not be possible to obtain constant regret in any case, it is an interesting question what could be gained by trying to adapt the satisfaction level towards the optimal mean reward. This is in particular interesting in view of the results for SAT-UCB$^+$, although currently provable regret bounds for this version of the algorithm are still missing.

A lesson to take from the MAB setting is that the savings from considering a satisficing instead of an optimizing objective –at least with respect to regret– is not that there are arms that need no exploration at all. Rather in the worst case (as always considered by notions of regret) one still has to explore all arms, however the amount of necessary exploration is now constant and independent of the horizon.

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

## A Complementary Experiments

In this section we report additional experiments that we performed in the following very easy *Setting 3*: There are 19 arms with mean reward 0 and one with mean reward 1. Here for any satisfaction level between 0 and 1 the task of satisficing is equivalent to learning the optimal arm. However knowing the satisfaction level gives some additional information that allows to learn faster. For the experiments $S = 0.5$ was chosen in the realizable case and $S = 1.1$ in the not realizable case.

As shown in Fig. 4b, in the realizable case SAT-UCB$^+$ and SAT-UCB work equally well in this very simple setting. Also UCB$_\alpha$ exhibits at least close to constant regret in Setting 3. In the not realizable case the two UCL variants perform best, cf. Figure 5a. As Figure 5b demonstrates, SAT-UCB$^+$ performs better the closer $S$ is chosen to $\mu_*$.

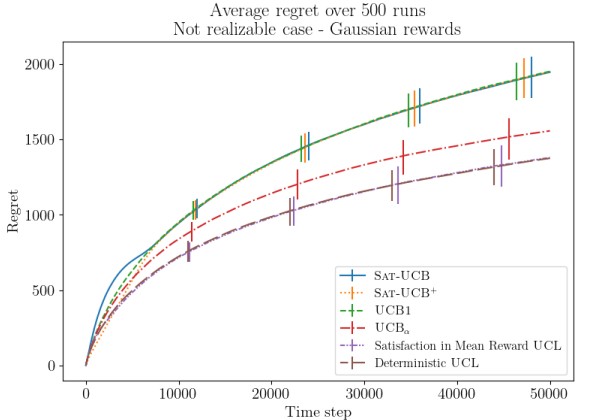
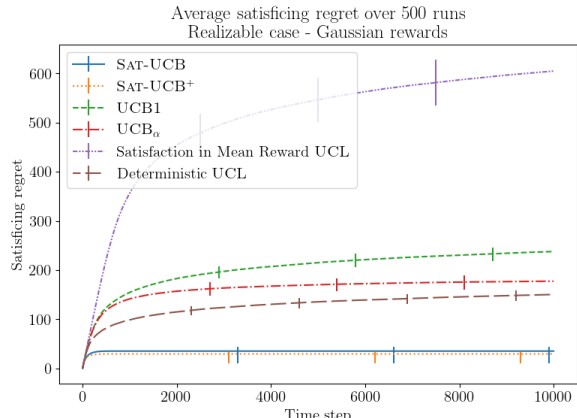

(a) Classic regret in the not realizable Setting 2 with $S = 1.1$.

(b) $S$-regret for the realizable case in Setting 3 with $S$=0.5.

Figure 4

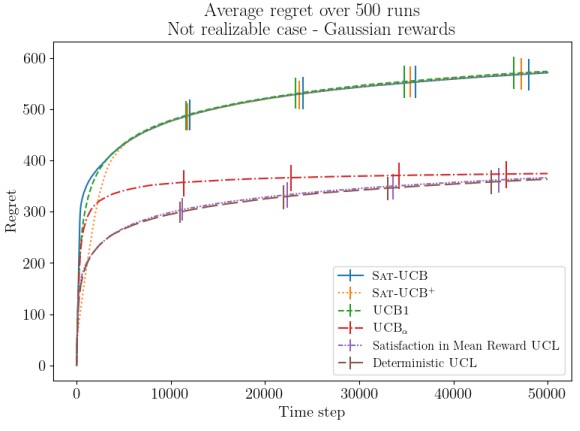
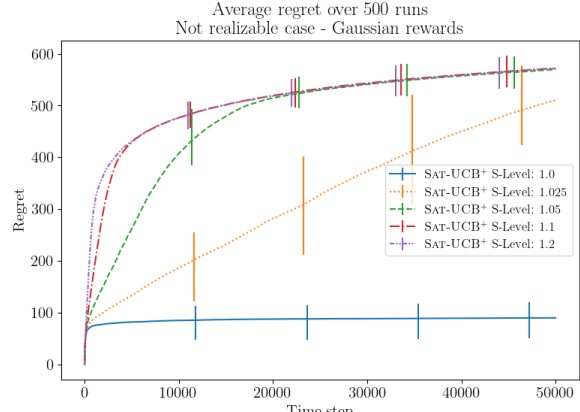

(a) Comparison of algorithms in Setting 3 with $S$=1.1.

(b) Comparison of different choices of $S$ for SAT-UCB$^+$.

Figure 5: Classic regret in the not realizable Setting 3.

