# OpenReview forum: "Regret Bounds for Satisficing in Multi-Armed Bandit Problems"
_TMLR — Rejected by TMLR_

### Review · Reviewer_1F3z · 2023-04-19

**Summary Of Contributions:**

The authors investigate a bandit problem where a threshold $S$ is given to the learner, and each play of an arm with mean larger than $S$ has no cost in the regret. $2$ cases emerge from this setting: if there exists such arm we can hope to get a constant regret (this is called the realizable case in the paper), while in the general case we can hope to perform as regular bandit algorithms. The contributions of the paper are:
* In Section 2 the authors analyze an algorithm in the case when the learner is ensured that the problem is realizable. The algorithm is (in spirit) greedy if at least one empirical mean is larger than $S$, and explores uniformly at random otherwise.
* In Section 3 they propose the Sat-UCB algorithm that combine 3 steps that depend on the value of the empirical means and UCB indices. In essence, the algorithm first consider the case when at least one arm is empirically satisfying, then the case where the UCB of at least one arm is satisfying, then the alternative (all UCBs are below $S$). They show that the regret is constant in the realizable case, and recovers the usual UCB guarantees in the general case.
* In Section 4, the authors present another version of Sat-UCB that aim to provide better empirical results by optimizing the exploitation phase. However, no guarantees are provided for this algorithm. They then provide some experiments to show to assess the performance of their algorithms on synthetic problems.

**Audience:**

No

**Claims And Evidence:**

Yes

**Requested Changes:**

* I think that the paper may be more interesting if an analysis of Sat-UCB++ was provided, and if this algorithm was motivated theoretically.
* It would be interesting to compare with the algorithms of Bubeck et al. (2013) in the realizable case.


**Strengths And Weaknesses:**

I think that the paper is clear and very easy to read. The setting is well presented and appropriately motivated in my opinion: in some applications of bandits it may make sense to assume that the user is happy with any arm that gives a large enough expected reward.

I am not really familiar with the works cited in the related work section, so I cannot really assess its completeness. I carefully read the paper from Bubeck (2013) for this review, and am of course familiar with UCB analysis, and even by considering those two works only the contribution of this paper seems rather weak in my opinion:
* Section 2 can be already entirely deduced from Bubeck (2013): their first policy perform a round-robin instead of random exploration but the analysis is exactly the same. I don't see the interest of dedicating a section to this algorithm, as it does not bring any insight regarding the other results. In my opinion, it should be summarized to a sentence saying that if the learner is ensured that one arm is strictly larger than the threshold then we can just use the algorithms/analysis of Bubeck (2023). Furthermore, their algorithms with potential functions have better guarantees.
* The analysis in Section 3 looks quite trivial to me. The algorithm distinguish cases where it can be greedy or not, and use UCB in the second case. It is quite obvious that in the case where the "greedy case" has low probability we just switch to UCB guarantees. Then, it is also clear that the analysis of Bubeck (2023) directly hold if these cases happen with high probability.

For these reasons, I don't see any particular insight in the results that are proposed. Even if they are correct, the results look largely trivial to me and I don't see a technical contribution in this paper. Hence, I am not sure that this paper is interesting for TMLR's audience.

---

> ### Author Response · Authors · 2023-04-25
> **Response to Reviewer 1F3z**
>
> We'd like to thank the reviewer for his/her time to take a close look at our paper and sharing his/her opinion with us.
>
> Please allow us to state our view on the submission before discussing detailed comments:
>
> Our paper presents a simple, yet effective way to approach a problem, which we think is interesting and $-$as also the reviewer concedes$-$ *"appropriately motivated"*. Our results present a rather complete picture of the satisficing regret: it is bounded if the sufficiency level is realizable, otherwise it is possible to obtain standard regret bounds. These results are new and have not been observed in the literature on the problem so far.
>
> We agree with the reviewer that the methods we use are simple and that the proofs for our results are straightforward, *once one knows how to approach them*. We do not think however that this makes the results of the paper obvious. Rather, as previous literature on the same setting shows, it is neither evident how to best approach the problem nor what the optimal bounds on the $S$-regret look like. Rather, it went e.g. unnoticed (by authors and reviewers alike) that the logarithmic lower bound stated by Reverdy et al (2017) and published in a renowned journal like *IEEE TAC* is actually wrong. Accordingly, we think it is unfair to call our results *"largely trivial"*.
>
> The aim of the paper was not about making a major technical contribution, but to provide a solution to the considered problem setting. That we were able to do so using simple means is in our view an asset, not a defect. At least, we do not see why it would be preferable to have a complicated algorithm with intricate analysis giving the same set of results.
>
> ### Detailed Comments
>
> **Section 2:**
> We agree that it is quite straightforward to adapt the proof of Bubeck et al (2013) to our setting. As the latter is however different from the standard setting, we thought it to be clearer to give a full proof (which is not that long anyway) than just a note that a reader would need to verify by him-/herself. Another motivation to include a full proof is that for correcting the claimed logarithmic lower bound of Reverdy et al (2017) it is preferable to have a complete and undisputable proof.
>
> We also have an adaptation of the refined analysis of Bubeck et al (2013) using potential functions, which we decided not to include in the paper (cf. remark at the end of Sec. 2 on p. 5), as the conditions under which this gives better bounds are not so clear.
>
> **Section 3:**
> While the overall idea of the analysis of the general algorithm in Section 3 is indeed simple, the details still have to be worked out. Actually, for the obvious adaptation of the same idea in the version of Sat-UCB+ we were not able to show a respective result. Again, it was not our goal to provide any technical novelties but to solve the problem of having an algorithm that has constant $S$-regret in the realizable case and at the same time is able to achieve logarithmic bounds otherwise. At least to us it was not clear in advance that it is possible to have such an algorithm.
>
> **Analysis for Sat-UCB+:**
> So far, we were not able to come up with a proof of constant regret bounds for Sat-UCB+ in the realizable case. Although it works well experimentally, we are not convinced that the exploration UCB does will be sufficient to obtain constant regret theoretically.
>
> **Experimental Comparison to Bubeck et al (2013):**
> We would be happy to add results of further experiments. However we are not sure which comparison algorithm the reviewer has in mind. The original algorithm of Bubeck et al (2013) takes as input the maximal average reward $\rho^*$ and the gap $\Delta$ to the second best arm. Then it basically uses $\rho^*-\Delta/2$ as satisficing level, that is, the algorithm works like our algorithm when choosing $S=\rho^*-\Delta/2$ (except that round robin is replaced by uniform exploration). Is the suggestion to compare to the algorithm that instead of using the realizable level $S$ itself rather takes $\rho^*-\Delta/2$ as input for the satisficing level?
>
> **Interest to TMLR community:**
> In our opinion we provide a complete solution to a natural problem which we think is in the scope of TMLR. We do not see why the audience of TMLR shouldn't be interested just because our approach is a simple and elegant application of existing results. Moreover, we are also convinced that the correction of  wrong claims in the literature is a service appreciated by the community.

---

> > ### Comment · Reviewer_1F3z · 2023-04-27
> > **Response to response**
> >
> > Thank you for your detailed answers and for expressing your opinion on several of my remarks. I would like to answer on some points:
> >
> > * to be clear, I am in no way advocating the fact that the solution of a problem should be complicated to make a paper interesting. However, I believe that this paper do not bring more insights than what was already discovered in Bubeck et al. (2013) (i.e the fact that no regret is achievable if it is known that the means are larger than a threshold). Providing the best mean and a lower bound on the gap" is conceptually equivalent to providing the threshold $S$. I also stick with the opinion that showing that the round-robin part can be replaced by UCB (and that in the worst case we get the regret of UCB) is something trivial.
> >
> > * Section 2/Section 3/Sat-UCB+ : my opinion remains unchanged but your comments are insightful, thank you. A precision: the refined analysis with potential functions gives better bounds when the best arm is only slightly above the threshold, by obtaining $\frac{\log(1/\Delta_\star^S)}{\Delta_i}$ instead of $\frac{1}{\Delta_\star^S}$.
> >
> > * Experiments: to use the algorithm from Bubeck et al. the threshold they use should simply be set to $S$. Furthermore, to include the possibility that $\mu^\star=S$ it is possible to also use Algorithm 1 from https://arxiv.org/pdf/1602.07182.pdf . I discovered after my review that this paper was also tackling this problem, it should be included in the references of the paper.

---

### Review · Reviewer_AVEa · 2023-04-30

**Summary Of Contributions:**

This study examines multi-armed bandit algorithms designed to minimize satisficing regret, which is defined as the deviation from a satisfaction level $S$, in contrast to conventional regret based on cumulative rewards. The authors present algorithms that depend on whether the expected reward of the best arm exceeds the satisfaction level $S$, and they establish upper bounds for their proposed algorithms. Finally, the authors validate the effectiveness of their proposed algorithms through simulation studies.


**Audience:**

Yes

**Broader Impact Concerns:**

No concerns.

**Claims And Evidence:**

Yes

**Requested Changes:**

The presentation is well-written, but there is room for improvement in some of the mathematical definitions and notations used. For instance:
- Could you clarify the spaces of $r_t$ and $\mu_i$?
- Is the number of time steps finite or infinite? What is the value of $T$?
- What do the symbols $[[\ \ \ ]]$ and $\sim$ (page 5)?
While I was able to guess their meanings, providing more detailed definitions may improve the readability of the paper.

Furthermore, I would appreciate more detailed explanations for certain sentences, such as the definition of sub-Gaussian random variables and the description of ``standard lower bounds'' on page 5. Including a citation would improve the clarity of this study.

**Strengths And Weaknesses:**

I believe that there are the following strengths and weaknesses.

Strengths:
- The problem setting is novel and has potential practical applications in the industry.
- The proposed algorithms are accompanied by theoretical guarantees.

Weaknesses:
- The explanations provided are somewhat coarse. More detailed descriptions could enhance the readability of the paper. (See Requested Changes below).
- It would be helpful to derive specific lower bounds for this problem. A discussion on the tightness of the upper bounds may also enhance the presentation.

Overall, while the paper offers contributions in both the problem setting and algorithms, there is room for improvement in explaining the results.

---

### Review · Reviewer_wJTK · 2023-05-03

**Summary Of Contributions:**

The paper analyzes a specific bandit setting in which the goal is to select an arm whose expected value is above a given threshold. The performances in this setting are evaluated using a different notion of regret than usual due to its characteristics. The authors propose two algorithms for the case in which the threshold is realizable and when not (i.e., when at least one arm has the expected value larger than the threshold). They show the regret bound for such algorithms. Finally, they propose a heuristic algorithm whose empirical performances are good in both realizable and non-realizable cases.


**Audience:**

Yes

**Broader Impact Concerns:**

I do not think such a theoretical work has a direct impact from an ethical point of view.

**Claims And Evidence:**

Yes

**Requested Changes:**

1) ad the definition of subgaussian and introduce the sub-gaussianity parameter.
2) specify that the expected value is over the stochasticity of the policy
3) "However, ..." This comment is not clear. Are you referring to a MAB with all the arms with the same expected value?
4) I would not make conjectures unless they are supported, for instance, by empirical results.
5) The writing can be improved. For instance, the second remark after Theorem 1 is not properly formulated. I suggest you to rewrite this paragraph.
6) "Bubeck ..." It is not clear to me how this remark relates to your work, Does it have the same regret bounds as yours?
7) The decomposition in Equation (6) is not clear. Please explain the rationale behind it. Maybe, the third term should contain an \exist symbol.
8) Sections 4.1 and 4.2 It is unclear to me the choice of including a new algorithm in the experimental part. I think this should be included as a contribution, stating that it is a heuristic algorithm without known regret bounds.
9) what do you mean by "very simple"
10) Why did you not use Thompson Sampling as a further baseline?
11) How did you compute the confidence intervals? Are they empirical estimates of the quantiles, or are you using a specific assumption on the mean value?
12)  From the figures, it seems that there is no statistical evidence that your method can provide a smaller S-regret. Is it true?





**Strengths And Weaknesses:**

The setting is clear. The algorithms are properly presented and analyzed.

Even if the motivation is sound, I think that being more specific and showing a specific application of the framework would strengthen the motivations behind this work.

Overall, I think the paper would require some rewriting since some concepts are sometimes not expressed linearly.

The experimental results are not convincing due to the wide confidence intervals provided.

---

### Review · Reviewer_LoTc · 2023-05-14

**Summary Of Contributions:**

**Summary:**

This paper considers the multi-armed bandit problem with the satisficing objective. In this problem, the learner pursues an arm whose reward is no smaller than a given satisfaction level $S$, instead of the optimal arm. The authors give algorithms and analysis for both the realizable setting (where a satisficing arm exists), and the general setting (where a satisficing arm may not exist). With a newly defined notion of satisficing regret, the authors provide a constant satisficing regret bound when there is a satisficing arm, and recover the standard logarithmic regret bound when a satisficing arm does not exist. Experimental results demonstrate that the proposed algorithm outperforms standard algorithms not only in the satisficing setting, but also in the classic bandit setting.


**Audience:**

Yes

**Broader Impact Concerns:**

I believe that this paper does not have ethical implications that will cause negative impacts.

**Claims And Evidence:**

Yes

**Requested Changes:**

Please see the weaknesses above.

**Strengths And Weaknesses:**

**Strengths:**

1.	The satisficing objective in multi-armed bandit is interesting and well-motivated.
2.	This paper is well-written and easy to follow. The idea of seeking for satisficing arms is clear and well executed. The analysis and results match intuition.
3.	The authors design algorithms and provide regret bounds for the settings where the learner knows the existence of a satisficing arm and does not. When there is a satisficing arm, the proposed algorithm achieves a constant regret bound. When there is no satisficing arm, the proposed algorithm re-establishes the classic logarithmic regret bound.
4.	The authors also conduct empirical evaluations to show the performance superiority of their algorithm over existing algorithms in both the satisficing setting and classic bandit setting.

**Weaknesses:**

1.	It would be better to provide more intuition behind algorithm design for the realizable and general settings.
2.	The theoretical results, e.g., Theorems 1-3, lack sufficient discussion and comments. It would be better to add more discussion following the theoretical results to give some insight and comparison with existing results.
3.	It seems that the literature review part lacks the discussion and comparison with prior threshold bandit works, e.g.,
Locatelli et al. An optimal algorithm for the Thresholding Bandit Problem. ICML 2016.

The notions of satisficing and threshold in multi-arm bandit should be very related. I suggest to include prior threshold bandit works in the literature review part.

**Overall Review:**

Overall, I think the satisficing multi-armed bandit problem is interesting and well-motivated. This paper develops algorithms and analysis for both the realizable setting and the general setting, and is well executed. The regret bounds and empirical results show that the proposed algorithm enjoys a constant regret in the satisficing setting, and also outperforms existing algorithms in the classic bandit setting.

---

### Decision · Action_Editors · 2023-06-04

**Recommendation:** Reject

**Comment:**

This paper studies satisficing bandits, a class of bandit problems with a threshold $S$ as an input. When the mean reward of arm $a$ is above the threshold, $\mu_a \geq S$, the regret for pulling it is zero. When it is not, the regret is $S - \mu_a$. The authors propose an algorithm whose regret is constant when a satisficing arm exists, its mean reward is above $S$; and logarithmic otherwise.

All reviewers agree that this paper would be a good match for TMLR. However, the current version needs a major revision to address many comments of the reviewers. The authors promised to do this on May 21. However, no revision was made, and therefore the paper is rejected for now.

**Audience:**

All reviewers agree that this paper would appeal to the general bandit audience.

**Claims And Evidence:**

The claims are not completely supported and this is one reason for rejection:

* Insufficient intuition behind algorithm designs and regret bounds.

* No lower bound. Therefore, the derived upper bounds may not be tight.

* Overlapping error bars in experiments. Therefore, statistical significance of some empirical results cannot be judged.

**Resubmission Of Major Revision:**

The authors may consider submitting a major revision at a later time.

---

> ### Author Response · Authors · 2023-06-06
> **Suprised by decision**
>
> To whom it may concern:
>
> About two weeks ago we were asked to prepare a revision of our paper by one of the reviewers. We have promised to do so and were about to finish this revision and submit it in the course of this week. Accordingly, we are quite surprised that now the paper has been rejected before we had a chance to submit our revision. It would have been nice if it we had been given a deadline in advance or if the action editor had checked with us if/when we were to submit the revision. In any case, we find it a bit strange that we are first asked for a revision, in which we have already put some efforts, and then we get rejected without any further notice.